# Evaluation of Temporal Change in IR Test Collections

## ABSTRACT

Information retrieval systems have been evaluated using the Cranfield paradigm for many years. This paradigm allows a systematic, fair, and reproducible evaluation of different retrieval methods in fixed experimental environments. However, real-world retrieval systems must cope with dynamic environments and temporal changes that affect the document collection, topical trends, and the individual user's perception of what is considered relevant. Yet, the temporal dimension in IR evaluations is still little studied.

To this end, this work investigates how the temporal generalizability of effectiveness evaluations can be assessed. As a conceptual model we generalize Cranfield type experiments to the temporal context by classifying the change in the essential components according to the operations of persistent storage known as CRUD. From the theoretical possible changes different evaluation scenarios are emerging and it is outlined what they imply. Based on these scenarios, renowned state-of-the-art retrieval systems are tested and it is investigated how the retrieval effectiveness changes on different levels of granularity.

We show that the proposed measures can be well adapted to describe the changes in the retrieval results. The experiments conducted confirm that the retrieval effectiveness strongly depends on the evaluation scenario investigated. We find that not only the average retrieval performance of single systems but also the relative system performance are strongly affected by the components that change and to what extent these components changed.

## CCS CONCEPTS

• **Information systems** → **Retrieval effectiveness**; *Novelty in information retrieval*; Temporal data; **Retrieval effectiveness**.

## KEYWORDS

Longitudinal Evaluation, Continuous Evaluation, Reproducibility

**ACM Reference Format:**
Anonymous Author(s). 2018. Evaluation of Temporal Change in IR Test Collections. In *Proceedings of Make sure to enter the correct conference title from your rights confirmation emai (Conference acronym 'XX)*. ACM, New York, NY, USA, 11 pages. https://doi.org/XXXXXXX.XXXXXXX

## 1 INTRODUCTION

Information Retrieval (IR) systems are exposed to constant change. The searched document collection evolves as new documents are added, removed, or updated [6, 19, 22]; the users always encounter

new information needs [17, 33, 45], and even the relevance is not static since information become outdated or opinions may change [12, 46]. In stark contrast, most IR experiments ignore the temporal dimension by only relying on snapshots or short time frames. By that, in test collection evaluations all temporal changes are abstracted, and their influence on the effectiveness is minimized. Multiple sources suggest that IR experiments based on test collections are not temporarily persistent [18, 24, 44]. Although some evolving dynamic test collections are available that span across more than one point in time, we identify the temporal dimension in IR evaluations as under-studied.

To investigate temporal dynamics in IR we focus on the question: How can the impact of temporal changes in the evaluation setup on the retrieval results be quantified? Therefore, describing the changes in the retrieval setup and measuring their impact on the effectiveness are the primary concerns. We focus on test collection evaluations as a starting point and address changes in documents and relevance labels. We propose to classify the changes in the different components of Cranfield test collections by the create, update and delete operation of CRUD as high level differentiation. Further, to investigate how changed effectiveness can be quantified different measures that are established in reproducibility evaluations are employed. To initially validate the proposed methodology, in the experimental evaluation we repeatedly evaluate five state-of-the-art IR systems in controlled evolved experimental setups based on the three established test collections: TripClick [39], TREC-COVID [48], and LongEval [2]. These test collections cover a range of temporal changes as highlighted in Fig. 1. It is shown how the adapted measures describe how the initial effectiveness measured (at $t_0$) relates to the effectiveness measured at a later point in time ($t_n$). Different aspects of changing effectiveness, independent of relevance, on the topic level and with focus on the system effect, are provided. This allows to set the established systems into context so that new insights about them can be achieved.

Since changes are unavoidable over time, we see great benefits in reintroducing temporal dynamics into test collection evaluations to learn about both, systems and test collections. Investigating temporal changes should help to improve the understanding of retrieval systems beyond their (relative) effectiveness by providing strategies to research how systems behave in specific situations. Investigating temporal changes can contribute to researching the reusability of test collections and emphasize the influence of the point of creation. Further, it can contribute to the field of test collection maintenance and to ensure reliably fair evaluations. Therefore, the proposed methodology, provides a more holistic understanding of IR evaluations in its temporal context and how the results depend on temporal effects.

In summary, the core contributions of this work are:

- Search scenarios are formally defined so that temporal changes can be systematically described. This formalization also serves as a classification schema to design longitudinal experiments.

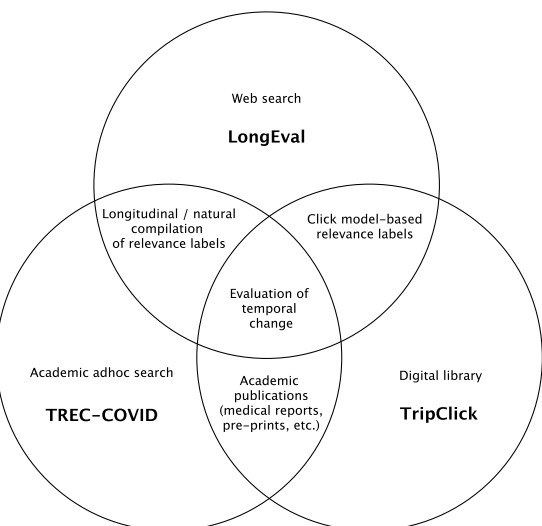

**Figure 1: Overview of the shared concepts between the test collections that stem from different evolving environments.**

- A comprehensive set of measures is introduced to systematically examine temporal influences on the retrieval effectiveness.
- The proposed methodology is tested in a comparative evaluation study with five state-of-the-art systems on three test collections covering different search scenarios.
- We finally discuss the methodology and results in the broader context and under different assumptions on longitudinal evaluations of retrieval effectiveness, which outlines directions for future work.

To facilitate reproducibility and improve transparency, we make all systems and evaluation code publicly available. [1]

## 2 RELATED WORK

To investigate the temporal relation between sub-collections, González-Sáez [21] compared the retrieval effectiveness achieved on consecutive sub-collections to random sub-collections. The results on three datasets showed that the effectiveness from temporal sub-collections varies stronger and is lower in general. Further, the variance between per-topic effectiveness' grows with increasing time between results. This indicates that the temporal dimension influences the experimental setup specifically, which leads to novel effects, different to cross test collection comparisons.

Only few works investigate the influence of temporal changes in IR evaluations directly. Soboroff [44] researched the reliability of evaluations in relation to changing documents in a web collection. He used the bpref measure due to its robustness against incomplete qrels and outlines how test collections can be maintained. Tonon et al. [47] put this into practice with the "evaluation as a service" methodology in which test collections are expanded over time to ensure a fair and valid evaluation. It is measured how well

---

[1]Will be added after review.

**Table 1: Exemplary evaluation scenarios indicated by the component and operation of change.**

|  | CREATE | UPDATE | DELETE |
|---|---|---|---|
| D'TQ | Extension of document collection | Document content changed (e.g., online news articles, or websites) | Documents removed (e.g., due to licensing issues) |
| DT'Q | New queries/topics (like current topics of interest) | Changed (head) queries from user logs (e.g., changed popularity) | Removed topics (due to missing interest or inappropriateness) |
| DTQ' | Added new relevance labels (from old or new assessors) | Assessors changed their mind; new judgment guidelines | Relevance labels removed (due to low inter-rater agreement) |

a current version of a test collection is suited to evaluate a system in comparison to prior evaluations and if updating the collection would be worth the needed effort. In this vein, Sáez et al. [42] apply evaluations in a similar setting by relating the measured effectiveness to a pivot system that is evaluated at the same point in time. They group the evaluation settings based on changing systems and environments. Their work focuses on the setting when the system, as well as the environment, changes. In contrast, conventional evaluations compare changed systems only and this work focuses on the setting where the environment is changing. To make the evolving effectiveness comparable over time, additionally to the pivot methodology, a projection strategy and grain comparisons are proposed [21]. Beyond test collection-based evaluations, Jensen et al. [25] repeatedly evaluate IR systems in a dynamic environment and investigate the needed query sample size for this evaluation setting.

Recently, the LongEval shared task [3] focused on the temporal persistence of retrieval systems by measuring the relative change in nDCG over different points in time, later described as $\mathcal{R}_e\Delta$. The LongEval test collection [2] was created specifically for this task, covering three points in time.

To relate the changes in the retrieval results to the changes in the collections they need to be quantified. While the proposed classification organizes the changes, we only superficially quantify them. In contrast to measuring the influence on the retrieval results more work is dedicated to measuring the changes in datasets and test collections [12, 17, 19, 22, 33, 45, 46]. Bar-Ilan [6], for example, investigates change in the web and proposes more detailed measures to quantify these changes.

## 3 SPECIFICATION OF THE EVOLVING EVALUATION ENVIRONMENT

The components necessary for evaluating IR systems are, while highly abstracted, provided by reusable test collections created following the Cranfield paradigm [13]. Due to the availability and

cost-efficient re-usability after initial creation, test collection-based evaluation experiments are the de facto standard in academic IR evaluations [10, 16, 32]. While such test collections provide great benefits for system-centered IR evaluations, they are only abstractions of the IR problem, neglecting most dynamics influencing the system in a real-world setting [23].

The evaluation setting needs to be formally defined to investigate how changes over time influence the retrieval effectiveness, so that measured changes in the evaluation setting can be attributed precisely and effects can be isolated systematically. Sáez et al. [42] describe the evaluation setting initially as the *Evaluation Environment (EE)* comprising the entirety of components needed for evaluation. While this EE evolves over time into a changed state denoted as EE', these changes influence the retrieval effectiveness [26].

Reintroducing temporal changes in test collections, for example through *evolving* test collections [44], the experimental setup can be represented as an evolving EE consisting of **D**ocuments, **T**opics, **Q**rels (DTQ). For a high-level classification, to distinguish how the components change, we align the changes to the basic operations of persistent storage CREATE, UPDATE, and DELETE known as CRUD [34]. Each operation has different implications on the effectiveness evaluations that are outlined in Tab. 1. Classifying the changes with CRUD provides an accessible differentiation without a content-related comparison.

This formal representation allows to allocate changes in the EE and is the foundation to quantify how these changes affect retrieval results. Changes in these components can be directly related to real scenarios IR systems are exposed to and, therefore, provide valuable insights into their capabilities if evaluated with these considerations. The changes in the EE are not exclusive to single components but rather often affect multiple components at once. For example, documents are added while different queries are issued simultaneously. Therefore, for some scenarios, changes can be investigated on a single component for others, it is necessary to consider multiple components at once. Additional effects from interactions between changes in multiple components can arise. Especially if, regarding influences on the effectiveness, it may be necessary to investigate changes in the documents or topics in conjunction with the related changes in the qrels. If new topics or documents are added to the EE (DT'Q and D'TQ), without also adding relevance labels they can not be assessed without new relevance labels.

## 4 MEASURING CHANGES IN EFFECTIVENESS

The temporal progression of the test collection is captured in evolving EEs. To measure how the effectiveness of an IR system changes across these EEs, runs created based on an evolved EE' are compared to the run of the initial EE. While a raw comparison of retrieval effectiveness superficially describes the change in effectiveness, it is highly influenced by the changes in the test collection, resulting in a changed recall base. This makes the measured scores hardly comparable.

With a focus on isolated scenarios where the document collection is changing (D'TQ), we adapt the reproducibility measures proposed by Breuer et al. [8, 31] and implemented in the repro_eval toolkit [9] to the temporal setting. For more detailed descriptions

and investigations of the measures, we refer to these works. On a high level, the *Rank Biased Overlap (RBO)* [49] compares two rankings of documents directly. In this case, the ranking $r$, produced by a system $S$ in the initial EE ($r_S^{EE}$) is compared to the ranking produced in the progressed EE' ($r_S^{EE'}$). Thereby, the change is assessed on the document level, independent of the relevance labels. Compared to Kendall's $\tau$ [27], the RBO weights higher-ranked documents as more important. It is defined as reproduced from Breuer et al. [8] and adapted to the temporal formalization:

$$\text{RBO}_j(r_S^{EE}, r_S^{EE'}) = (1 - \phi) \sum_{i=1}^{\infty} \phi^{i-1} \cdot A_i,$$

$$\overline{\text{RBO}}(r_S^{EE}, r_S^{EE'}) = \frac{1}{n^{EE}} \sum_{j=1}^{n^{EE}} \text{RBO}_j(r_S^{EE}, r_S^{EE'}). \tag{1}$$

The RBO is calculated per topic $j$ and then averaged over all topics in the EE. The parameter $\phi$ is bound between 0 and 1 and adjusts the weighting of the rank. The smaller $\phi$ is chosen, the higher the top ranks are weighted. $A_i$ is the overlap between the two rankings up to rank $i$, which can be formalized as $|S_{:i}^{EE} \cap S_{:i}^{EE'}|$. Likewise, the $\overline{RBO}$ is the average of all topic-wise RBOs and summarizes the document-level similarity. Since the comparison is done on the topic level, it can only be applied to an evolved EE' in which the topics are the same. The higher the RBO is, the more similar the two document rankings are and the smaller should be the changes in effectiveness.

In contrast to the RBO, the *Root Mean Square Error (RMSE)* incorporates the relevance labels. The RMSE quantifies the error between the effectiveness scores of a measure $M$ per topic $j$ of a system $S$ at one EE compared to an evolved EE'. While the RBO directly compares the two rankings on the document level, the RMSE operates on the achieved effectiveness. This means that the RMSE does not account for changes in the ranking as long as the documents are equally relevant. The RMSE is applied to the D'TQ scenario, which means that the effectiveness of the EE' run is measured using the qrels set of the initial EE. Hence, the recall base is not changing, and the measured effectiveness can be compared. In this context, the RMSE is defined as:

$$\text{RMSE}\left(M^{EE}(S), M^{EE'}(S)\right) = \sqrt{\frac{1}{n^{EE}} \sum_{j=1}^{n^{EE}} \left(M_j^{EE}(S) - M_j^{EE'}(S)\right)^2}. \tag{2}$$

The effectiveness, measured on a topic level by a measure $M$, of the advanced system $S$ is determined based on the two EEs. The difference between the measured effectiveness is then directly compared on the topic level and squared to avoid compensation of a positive change through a negative change during averaging. Additionally, the RMSE puts more weight on larger differences through squaring them [8]. The higher the RMSE is, the larger the error between the two runs, and, therefore the larger the changes in effectiveness. Through this setup, the effect of the changing documents on the system is isolated. Therefore, a larger RMSE can be interpreted as a stronger influence of changing documents or relevance, depending on the experimental setup.

The described reproducibility measures focus on scenarios with changing documents. Further, a set of measures is adapted that are suitable to compare changes in highly dynamic EEs with changes in topics and qrels. These measures originate from replicability investigations where a changed test collection is used. As a first step towards measuring retrieval effectiveness over time in highly dynamic EEs Sáez et al. [41] propose the idea of *Result Deltas (R$\Delta$)*. In its simplest form, the environment $\mathcal{R}_e\Delta$ compares the same system in changing EEs. The $\mathcal{R}_e\Delta$ is used in LongEval [3] as:

$$\mathcal{R}_e\Delta = \frac{\overline{M^{EE}(S)} - \overline{M^{EE'}(S)}}{\overline{M^{EE}(S)}}. \qquad (3)$$

Given a retrieval system $S$, the delta between the average effectiveness based on a retrieval measure $M$ in an EE and an evolved EE' is calculated and normalized by the measured effectiveness on the initial EE. Ideally, this measure describes how the overall effectiveness of the system changes between the two EEs. However, if the evolving EEs include changes in the qrels, this measure is directly impacted by the different recall bases what may leade to skewed results and limit the applicability of the measure.

Breuer et al. [8] propose the *Delta Relative Improvement ($\Delta$RI)* to investigate the replicability of a system in a different experimental setup, the evolved EE in this case. These measures can be applied to highly dynamic EEs where all components may evolve (D'T'Q').

The $\Delta$RI can be seen as the reproducibility approach to the $\mathcal{R}_e\Delta$. In this context [8], it is defined as:

$$\mathrm{RI} = \frac{\overline{M^{EE}(S)} - \overline{M^{EE}(P)}}{\overline{M^{EE}(P)}}, \qquad \mathrm{RI'} = \frac{\overline{M^{EE'}(S)} - \overline{M^{EE'}(P)}}{\overline{M^{EE'}(P)}}, \qquad (4)$$

$$\Delta\mathrm{RI} = \mathrm{RI} - \mathrm{RI'}.$$

First, the *Relative Improvement (RI)* is calculated as the per-topic delta between the measured effectiveness given a measure $M$ for the experimental system $S$ and the pivot system $P$, separately on EE and EE'. Finally, the delta between the two RIs is taken. Since the difference in effectiveness between $S$ and $P$ is not determined across different EEs, changing topics remains possible. In this setting, the $\Delta$RI can be interpreted as the change in effectiveness in relation to the pivot system between the initial EE and the evolved EE'. A perfect reproduction, equivalent to consistent effectiveness, would yield a $\Delta$RI of 0. If $\Delta$RI > 0, the improvement over the pivot system is decreased; if $\Delta$RI < 0, it is increased. Intuitively, the $\Delta$RI is strongly related to the $\mathcal{R}_e\Delta$ as they both pursue to measure the change in effectiveness. While the $\mathcal{R}_e\Delta$ directly relates the results of two points in time, the $\Delta$RI compares the results in relation to the pivot system. This is essentially what Sáez et al. [42] propose as pivot evaluation.

## 5 EXPERIMENTAL SETUP

The proposed measures are tested on different evaluation scenarios investigating the influence of changes in the document collection. Three test collections are used: TripClick, TREC-COVID, and LongEval, and their dynamics are summarized in Table 2. The changes in the document component (D'TQ) are investigated through RBO and RMSE. Further, changes in the document component with the related changes in the qrels component (D'TQ')

are described by the $\mathcal{R}_e\Delta$ and $\Delta$RI. All measures that rely on the effectiveness are instantiated with P@10, bpref, and nDCG. Since the focus is on changes in the document component, a set of topics is used that is common to all EEs in one dataset.

In the first scenario, CREATE changes are explored. The scenario shows strongly controlled changes spanning longer periods of time. The scenario is based on the TripClick test collection is used [39] that consists of over 1.5 million scientific medical publications and user interactions. The topics are separated based on the frequencies they were issued into head, torso, and tail, of which we used the 1,175 topics of the test set based on the DCTR [15] click model. While this test collection is not intended to be dynamic, we simulated evolving EEs by separating the document corpus by the publication dates of the documents into three equally sized sub-collections spanning multiple years. The intuition is that further publications become available over time. In conjunction with the additional documents, the corresponding qrels are also added to the evolved EE. This simulated scenario resembles change across a longer period but with far fewer types of change. Since documents are only created and not deleted or updated, an append-only collection is simulated as it may be found in archival systems or digital libraries.

The second scenario coveres mainly CREATE changes and additionally UPDATE and DELETE changes. It is less controlled, investigates multiple types of changes, and shows stronger changes over shorter periods. These changes are found in the TREC-COVID test collection [48]. Like the TripClick test collection, it contains scientific publications with a focus on COVID-19. This test collection was constructed over a shorter time frame in five rounds in which the documents changed, topics were added, and the relevance was assessed. Therefore, it can also be seen as a naturally evolving test collection. However, the COVID-19 pandemic was an exceptional example, and as such, search was influenced by severe dynamics. This is reflected in the test collection by drastic changes in the qrels and documents. However, similar dense dynamics can also be observed in other search scenarios such as social media, financial markets, or news. Compared to the other two test collections, TREC-COVID has smaller but deeper pools.

The third scenario shows all three types of changes: CREATE, UPDATE, and DELETE. It already starts with a large collection in the beginning and shows moderate changes over timeframes of months. For this scenario, the LongEval test collection is used. It was introduced in the LongEval CLEF lab in 2023 [20][2] and it is a naturally evolving test collection made for longitudinal evaluations in IR. The test collection is available in French but with additional English machine translations, which are used for this evaluation. Since it consists of three sub-collections from three consecutive time spans of one month each, with an additional month gap between the last sub-collections, it spans a moderate period of time. The test collection originates from the web domain and, therefore, the EEs are highly dynamic yet still comparable [21]. Over 1.5 million websites in total and up to 910 queries per sub-collection are contained in the test collection. The qrels are constructed with a cascading click model based on logged user interactions [11]. Thus, the test collection relies on shallow pools with many topics.

---

[2]https://clef-longeval.github.io/

**Table 2: High-level overview of some statistics of the different components in the test collections across the EEs. The total number of documents, topics and qrels is reported and additionally the percentage (in parentheses) changed compared to the total number of the first EE ($t_0$). The changes in the document component compared to $t_0$ are additionally reported based on the CREATE, UPDATE, and DELETE operations. The update changes are determined by comparing the string length for documents with the same ID (URL in the case of LongEval).**

|  |  | Documents (D) | | | | Topic (T) | | Qrels (Q) | |
|---|---|---|---|---|---|---|---|---|---|
|  |  | total | % | CREATE | UPDATE | DELETE | total | % | total | % |
| TripClick | $t_0$ | 565,737 | | | | | 1,175 | | 14,334 | |
| | $t_1$ | 1,085,094 | (92%) | 519,357 | 0 | 0 | 1,175 | (0%) | 37,710 | (163%) |
| | $t_2$ | 1,510,743 | (167%) | 945,006 | 0 | 0 | 1,175 | (0%) | 67,943 | (374%) |
| TREC-COVID | $t_0$ | 51,045 | | | | | 30 | | 8,691 | |
| | $t_1$ | 59,851 | (17%) | 8,828 | 22 | 188 | 35 | (17%) | 10,293 | (18%) |
| | $t_2$ | 128,162 | (151%) | 75,562 | 7,251 | 8,468 | 40 | (33%) | 9,517 | (10%) |
| | $t_3$ | 157,817 | (209%) | 36,750 | 7,095 | 8,478 | 45 | (50%) | 7,298 | (-16%) |
| | $t_4$ | 191,175 | (275%) | 36,750 | 1,319 | 8,076 | 50 | (67%) | 9,779 | (13%) |
| LongEval | $t_0{}^3$ | 1,570,734 | | | | | 753 | | 2,037 | |
| | $t_1$ | 1,593,376 | (1%) | 40,859 | 196,682 | 18,217 | 860 | (14%) | 2,065 | (2%) |
| | $t_2$ | 1,081,334 | (31%) | 46,837 | 192,624 | 558,879 | 910 | (21%) | 2,056 | (1%) |

While these test collections describe different search scenarios, they also have multiple commonalities that are visualized in Fig. 1. The LongEval test collection relies on a click model to estimate the relevance of documents like the TripClick test collection. Both the TripClick test collection and the TREC-COVID test collection originate from the scientific domain the material collection consists of scientific papers. The TREC-COVID and LongEval test collections are both naturally evolving test collections, as they reflect real observed changes. These commonalities make the different scenarios comparable and enable a comprehensive discussion.

On the system side, we reproduced runs with five state-of-the-art retrieval systems. The systems were chosen to cover a variety of system architectures. No further adaption to evolving EEs or any fine-tuning to the test collections was performed to investigate the systems in an "off the shelf" state and improve comparability.

BM25 [40] is used as the pivot system and also as the first retrieval stage for most advanced systems. The advanced systems are BM25 with MonoT5 reranking [35, 38], BM25 with ColBERT reranking [28], an index, expanded with 10 additional queries generated with Doc2Query (d2q) [36, 37] and queried with BM25 and a Reciprocal Rank Fusion (RRF) [14] approach based on three runs from BM25 with Boolean query expansion [4], Divergence from Randomness (DFR) with parameter-free $\chi^2$ weighting [5] and PL2 [4]. The systems are implemented through PyTerrier [30] and Ranx [7], with the default parameters. Besides an RRF system based on established approaches, ColBERT and MonoT5 represent different systems that rely on large language models (LLMs) for re-ranking. The d2q system, in addition, represents a reversed approach by first enriching the documents through an LLM and then using BM25 for retrieval. The replicability measures were implemented through repro_eval [9], which is a dedicated reproducibility and replicability evaluation toolkit.

## 6 RESULTS

We note that the goal is not to assess the effectiveness of the systems but to learn how systems and collections change under evolving conditions. Ranking the systems from a "persistent point of view" would not befit the problem. Also, no single best measure can be determined since they cover different aspects of temporal change. The results are presented in Tab. 3. Additionally, the changes in the average retrieval performance (ARP) is visualized in Fig. 2.

The RBO is determined for rankings with a length of 100 documents and clearly decreases over time, as shown by the decreasing rank similarity over the EEs in Tab. 3. This effect is especially visible for the TripClick and TREC-COVID datasets with fewer pronounced types of change and vanishes for LongEval with more diverse changes. On TripClick, the similarity of the rankings is steadily decreasing for all systems. On TREC-COVID, an initially higher similarity drops fast and almost converges in the last EEs where nearly no similarity is present. A similar gradient can be seen for all systems. For LongEval, the results differ per system and vary less between EEs. The ranking at $t_2$ for the systems BM25, MonoT5, and d2q appears to be more similar to $t_0$ compared to $t_1$. ColBERT and MonoT5 have especially similar rankings in all EEs.

The error in effectiveness measured by the RMSE generally agrees with the ranking similarity described by the RBO across all instantiations. The RMSE increases with progressing EEs for almost all systems, however, more slowly if based on bpref. This shows a more indulgent behavior. While the RBO shows a strongly changed ranking for the TripClick test collection, the RMSE only indicates smaller errors. On this test collection the d2q system shows the highest error in bpref for both EEs. ColBERT has the lowest error in the TREC-COVID test collection. For LongEval, MonoT5 has a lower error than the other systems if the RMSE is instantiated with P@10 and nDCG, ColBERT has a low error in bpref.

The ARP for the three effectiveness measures P@10, bpref, and nDCG are reported for reference. Since they are calculated based

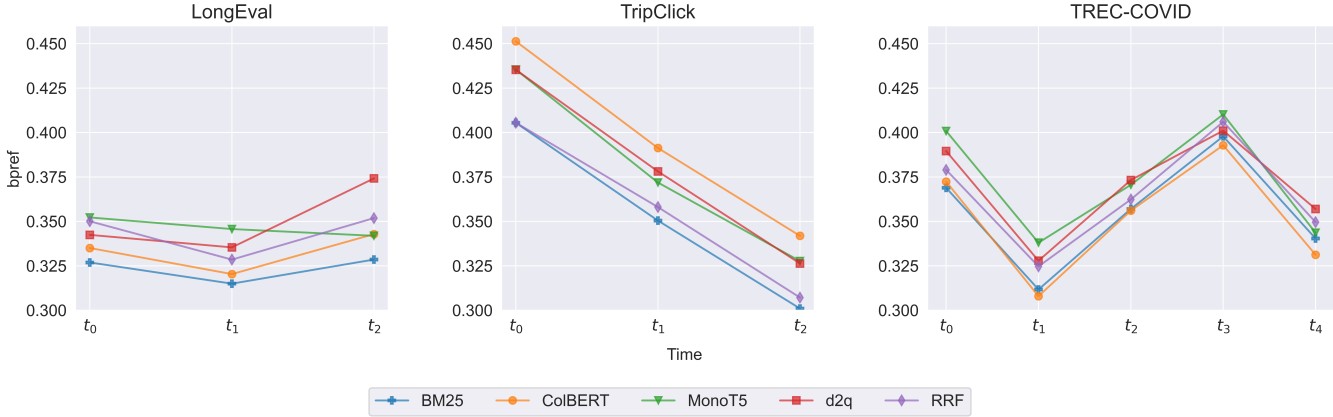

**Figure 2: Retrieval effectiveness of the different systems measured by bpref for the three test collections over multiple sub-collections. No strong agreement for a system ranking between test collections or sub-collections can be found.**

on different qrels, the scores might not be directly comparable. The ARP shows no explicit agreement across the different systems, test collections, or EEs. A slight trend toward MonoT5 and d2q being the best-performing systems can be observed, at least for LongEval and TripClick. For TREC-COVID, these systems are outperformed by RRF in the later EEs. The ARP measured with bpref for all test collections and sub-collections is visualized in Fig. 2. As displayed, it seems like the effectiveness varies across EEs. For LongEval, the effectiveness decreases at first and then increases again, while for TripClick, the effectiveness steadily decreases with the evolving EE. The TREC-COVID interpolation is highly varying and does not show a clear trend. While the differences between LongEval and TREC-COVID are similarly small, the effectiveness scores for the TripClick test collection have a more extensive range. The ranking of systems appears to be unstable across the different points in time on all test collections. The differences between the pivot systems (BM25 at $t_0$) and the experimental systems at the later EEs are tested for significance with a t-test, Bonferroni correction, and $\alpha = 0.05$. For the LongEval and TREC-COVID test collections only some results are significantly different. The TripClick test collection shows more significant differences.

The $\mathcal{R}_e\Delta$ and $\Delta$RI indicate how the effectiveness changes based on the raw effectiveness and measured in relation to a pivot system. Generally stronger changes are observed for the effectiveness measured by P@10 compared to the other measures. As observed before, the simulated EEs based on TripClick show clear patterns. The ARP increases for P@10 and nDCG and decreases for bpref. The $\mathcal{R}_e\Delta$ agrees with these effects largely and diverges from 0, indicating an increased or decreased effectiveness. The $\Delta$RI shows much more nuanced differences. Instantiated with P@10 it agrees with the $\mathcal{R}_e\Delta$ for ColBERT and MonoT5; for RRF and d2q, different changes are measured. Based on bpref for MonoT5 the measured changes turn into the opposite for $t_2$. For nDCG the agreement with the $\mathcal{R}_e\Delta$ appears to be reversed. In summary, based on the TripClick dataset, the $\Delta$RI shows only minor changes in effectiveness while the $\mathcal{R}_e\Delta$ agrees with the trends of the ARP.

On TREC-COVID, the effectiveness varies across EEs. This is also reflected in the $\mathcal{R}_e\Delta$, which again generally agrees with the ARP. In this comparison, the $\mathcal{R}_e\Delta$ instantiated with bpref shows the least agreement, especially for RRF and BM25. Like before on TripClick, the $\Delta$RI indicates less strong changes, especially if based on bpref but also for P@10 and nDCG. $\mathcal{R}_e\Delta$ and $\Delta$RI in comparison show rather an agreement across systems then instantiated measures on this test collection. Even less agreement on the direction of change appears. For example, for ColBERT at $t_3$, the $\mathcal{R}_e\Delta$ shows a slightly increased effectiveness while $\Delta$RI indicates a decreased effectiveness.

On the LongEval test collection, the ARP slightly impairs first and then increases again beyond the initial EE. This is reflected in the $\mathcal{R}_e\Delta$ across all measures and systems. Similar to the observations before, the $\Delta$RI does not allways agree on the direction of change with the $\mathcal{R}_e\Delta$. Further the changes described by the $\Delta$RI appear to be smaller compared to the $\mathcal{R}_e\Delta$.

With additional components that change (D'TQ compared to D'TQ') in the investigated scenarios and with stronger overlapping types of change the results variate stronger and patterns become less visible.

## 7 DISCUSSION

The results of the experimental evaluation show how the measures describe different aspects of temporal changes in a variety of scenarios. Notably, we have to distinguish between (1) the evaluation of retrieval systems and (2) the assessment of the data collection as the evaluation tool. Suppose that we know the system ranking in advance with high confidence and the test collection results in a different ranking at a later point in time. In that case, the test collection would require additional curation to be considered as an integrated collection. On the other hand, the temporal changes in the data help to evaluate the robustness of retrieval systems and how they adapt in a dynamic environment.

To further interpret the results and deepen the understanding of the proposed methodology, we discuss the observations along central aspects of temporal changes in IR evaluations. The different

**Table 3: Experimental results of the different retrieval systems across test collections and EEs. The left side of the table describes results solely based on changes in the document component (D'TQ), and the right side additionally with the changes in the qrels (D'TQ'). The $\Delta RI$ relies on BM25 as a pivot system, and therefore, no values can be calculated for this system. $t_0$ is always used as the reference EE and, therefore, the ideal values are displayed in the $t_0$ rows. Cells per test collection and measure are highlighted darker if they indicate less change compared to $t_0$. Thus, on a high level, rows with darker colors indicate systems that change less. For a fine-grained analysis, the results of one measure can be compared across its different instantiations. Statistically significant differences in the ARP from BM25 are marked with an asterisk (*).**

| | | | D'TQ RBO@100 / RBO | D'TQ P@10 | D'TQ bpref / RMSE | D'TQ nDCG | D'TQ' P@10 ARP | $\mathcal{R}_e\Delta$ | $\Delta$RI | D'TQ' bpref ARP | $\mathcal{R}_e\Delta$ | $\Delta$RI | D'TQ' nDCG ARP | $\mathcal{R}_e\Delta$ | $\Delta$RI |
|---|---|---|---|---|---|---|---|---|---|---|---|---|---|---|---|
| **TripClick** | | | | | | | | | | | | | | | |
| CREATE | BM25 | $t_0$ | 1 | 0 | 0 | 0 | 0.081 | 0 | - | 0.405 | 0 | - | 0.280 | 0 | - |
| | | $t_1$ | 0.583 | 0.066 | 0.102 | 0.081 | 0.111 | -0.377 | - | 0.350 | 0.136 | - | 0.323 | -0.155 | - |
| | | $t_2$ | 0.448 | 0.084 | 0.131 | 0.107 | 0.123 | -0.522 | - | 0.301 | 0.258 | - | 0.334 | -0.193 | - |
| | ColBERT | $t_0$ | 1 | 0 | 0 | 0 | 0.096* | 0 | 0 | 0.451* | 0 | 0 | 0.298* | 0 | 0 |
| | | $t_1$ | 0.577 | 0.076 | 0.107 | 0.094 | 0.130* | -0.356 | 0.018 | 0.391* | 0.133 | -0.004 | 0.337* | -0.130 | 0.023 |
| | | $t_2$ | 0.450 | 0.099 | 0.148 | 0.119 | 0.142* | -0.481 | 0.031 | 0.342* | 0.242 | -0.023 | 0.350* | -0.175 | 0.017 |
| | MonoT5 | $t_0$ | 1 | 0 | 0 | 0 | 0.090* | 0 | 0 | 0.435* | 0 | 0 | 0.291 | 0 | 0 |
| | | $t_1$ | 0.582 | 0.075 | 0.100 | 0.093 | 0.121* | -0.354 | 0.018 | 0.372* | 0.146 | 0.013 | 0.334* | -0.148 | 0.006 |
| | | $t_2$ | 0.447 | 0.097 | 0.139 | 0.118 | 0.131 | -0.459 | 0.046 | 0.328* | 0.247 | -0.015 | 0.347* | -0.193 | 0.000 |
| | RRF | $t_0$ | 1 | 0 | 0 | 0 | 0.085* | 0 | 0 | 0.406 | 0 | 0 | 0.285 | 0 | 0 |
| | | $t_1$ | 0.552 | 0.074 | 0.109 | 0.093 | 0.113 | -0.333 | 0.033 | 0.358 | 0.117 | -0.022 | 0.326 | -0.146 | 0.008 |
| | | $t_2$ | 0.421 | 0.093 | 0.129 | 0.118 | 0.126 | -0.489 | 0.022 | 0.307* | 0.243 | -0.020 | 0.338* | -0.188 | 0.004 |
| | d2q | $t_0$ | 1 | 0 | 0 | 0 | 0.092* | 0 | 0 | 0.435* | 0 | 0 | 0.303* | 0 | 0 |
| | | $t_1$ | 0.469 | 0.073 | 0.169 | 0.103 | 0.128* | -0.382 | -0.004 | 0.378* | 0.131 | -0.005 | 0.349* | -0.153 | 0.002 |
| | | $t_2$ | 0.386 | 0.088 | 0.185 | 0.126 | 0.140* | -0.514 | 0.006 | 0.326* | 0.250 | -0.011 | 0.363* | -0.200 | -0.006 |
| **TREC-COVID** | | | | | | | | | | | | | | | |
| CREATE, UPDATE, DELETE | BM25 | $t_0$ | 1 | 0 | 0 | 0 | 0.430 | 0 | - | 0.369 | 0 | - | 0.421 | 0 | - |
| | | $t_1$ | 0.761 | 0.215 | 0.012 | 0.067 | 0.177 | 0.589 | - | 0.312 | 0.155 | - | 0.301 | 0.286 | - |
| | | $t_2$ | 0.317 | 0.354 | 0.075 | 0.193 | 0.263 | 0.388 | - | 0.357 | 0.032 | - | 0.334 | 0.207 | - |
| | | $t_3$ | 0.207 | 0.418 | 0.117 | 0.249 | 0.253 | 0.411 | - | 0.398 | -0.078 | - | 0.360 | 0.144 | - |
| | | $t_4$ | 0.177 | 0.427 | 0.149 | 0.280 | 0.233 | 0.457 | - | 0.340 | 0.077 | - | 0.309 | 0.265 | - |
| | ColBERT | $t_0$ | 1 | 0 | 0 | 0 | 0.407 | 0 | 0 | 0.372 | 0 | 0 | 0.389 | 0 | 0 |
| | | $t_1$ | 0.709 | 0.161 | 0.015 | 0.050 | 0.200 | 0.508 | -0.186 | 0.308 | 0.173 | 0.022 | 0.284 | 0.270 | -0.020 |
| | | $t_2$ | 0.235 | 0.333 | 0.083 | 0.155 | 0.247 | 0.393 | 0.009 | 0.356 | 0.044 | 0.012 | 0.321 | 0.176 | -0.036 |
| | | $t_3$ | 0.156 | 0.349 | 0.123 | 0.204 | 0.167* | 0.590 | 0.288 | 0.393 | -0.055 | 0.022 | 0.327 | 0.159 | 0.015 |
| | | $t_4$ | 0.136 | 0.357 | 0.151 | 0.233 | 0.163 | 0.598 | 0.246 | 0.331 | 0.110 | 0.036 | 0.298 | 0.233 | -0.040 |
| | MonoT5 | $t_0$ | 1 | 0 | 0 | 0 | 0.483 | 0 | 0 | 0.401 | 0 | 0 | 0.439 | 0 | 0 |
| | | $t_1$ | 0.761 | 0.188 | 0.014 | 0.068 | 0.220 | 0.545 | -0.121 | 0.338* | 0.157 | 0.002 | 0.319 | 0.274 | -0.017 |
| | | $t_2$ | 0.311 | 0.390 | 0.087 | 0.199 | 0.310 | 0.359 | -0.053 | 0.371 | 0.076 | 0.049 | 0.344 | 0.215 | 0.011 |
| | | $t_3$ | 0.190 | 0.454 | 0.134 | 0.248 | 0.237 | 0.510 | 0.190 | 0.410 | -0.023 | 0.056 | 0.363 | 0.172 | 0.034 |
| | | $t_4$ | 0.161 | 0.485 | 0.163 | 0.282 | 0.187 | 0.614 | 0.324 | 0.344 | 0.143 | 0.077 | 0.319 | 0.272 | 0.010 |
| | RRF | $t_0$ | 1 | 0 | 0 | 0 | 0.453 | 0 | 0 | 0.379 | 0 | 0 | 0.439* | 0 | 0 |
| | | $t_1$ | 0.729 | 0.245 | 0.019 | 0.070 | 0.190 | 0.581 | -0.021 | 0.325* | 0.143 | -0.014 | 0.316* | 0.279 | -0.009 |
| | | $t_2$ | 0.309 | 0.381 | 0.081 | 0.207 | 0.267 | 0.412 | 0.042 | 0.362 | 0.044 | 0.012 | 0.340 | 0.226 | 0.025 |
| | | $t_3$ | 0.191 | 0.433 | 0.123 | 0.261 | 0.277 | 0.390 | -0.038 | 0.406 | -0.071 | 0.007 | 0.373* | 0.150 | 0.007 |
| | | $t_4$ | 0.139 | 0.459 | 0.150 | 0.291 | 0.243 | 0.463 | 0.011 | 0.350 | 0.078 | 0.000 | 0.322* | 0.267 | 0.003 |
| | d2q | $t_0$ | 1 | 0 | 0 | 0 | 0.407 | 0 | 0 | 0.390* | 0 | 0 | 0.435 | 0 | 0 |
| | | $t_1$ | 0.502 | 0.187 | 0.025 | 0.072 | 0.200 | 0.508 | -0.186 | 0.328 | 0.158 | 0.004 | 0.317 | 0.271 | -0.021 |
| | | $t_2$ | 0.128 | 0.389 | 0.090 | 0.214 | 0.277 | 0.320 | -0.105 | 0.373 | 0.042 | 0.010 | 0.345 | 0.207 | 0.001 |
| | | $t_3$ | 0.083 | 0.418 | 0.131 | 0.265 | 0.183* | 0.549 | 0.222 | 0.401 | -0.030 | 0.048 | 0.341* | 0.217 | 0.088 |
| | | $t_4$ | 0.069 | 0.441 | 0.157 | 0.294 | 0.190 | 0.533 | 0.131 | 0.357 | 0.084 | 0.007 | 0.315 | 0.276 | 0.015 |
| **LongEval** | | | | | | | | | | | | | | | |
| CREATE, UPDATE, DELETE | BM25 | $t_0$ | 1 | 0 | 0 | 0 | 0.100 | 0 | - | 0.327 | 0 | - | 0.282 | 0 | - |
| | | $t_1$ | 0.594 | 0.062 | 0.129 | 0.141 | 0.086 | 0.137 | - | 0.315 | 0.036 | - | 0.273 | 0.033 | - |
| | | $t_2$ | 0.603 | 0.070 | 0.132 | 0.139 | 0.110 | -0.105 | - | 0.329 | -0.005 | - | 0.306 | -0.086 | - |
| | ColBERT | $t_0$ | 1 | 0 | 0 | 0 | 0.102 | 0 | 0 | 0.335 | 0 | 0 | 0.286 | 0 | 0 |
| | | $t_1$ | 0.593 | 0.065 | 0.137 | 0.141 | 0.092 | 0.095 | -0.049 | 0.320 | 0.044 | 0.008 | 0.274 | 0.042 | 0.010 |
| | | $t_2$ | 0.583 | 0.070 | 0.144 | 0.137 | 0.119 | -0.175 | -0.064 | 0.343 | -0.023 | -0.019 | 0.298 | -0.042 | 0.041 |
| | MonoT5 | $t_0$ | 1 | 0 | 0 | 0 | 0.113 | 0 | 0 | 0.352 | 0 | 0 | 0.309 | 0 | 0 |
| | | $t_1$ | 0.594 | 0.077 | 0.176 | 0.138 | 0.106 | 0.057 | -0.105 | 0.346 | 0.019 | -0.020 | 0.302 | 0.020 | -0.015 |
| | | $t_2$ | 0.602 | 0.075 | 0.181 | 0.135 | 0.123 | -0.093 | 0.012 | 0.342 | 0.029 | 0.037 | 0.311 | -0.009 | 0.077 |
| | RRF | $t_0$ | 1 | 0 | 0 | 0 | 0.106 | 0 | 0 | 0.350 | 0 | 0 | 0.292 | 0 | 0 |
| | | $t_1$ | 0.579 | 0.060 | 0.166 | 0.127 | 0.089 | 0.167 | 0.036 | 0.328 | 0.062 | 0.028 | 0.282 | 0.035 | 0.002 |
| | | $t_2$ | 0.544 | 0.071 | 0.165 | 0.129 | 0.121 | -0.136 | -0.030 | 0.352* | -0.005 | -0.000 | 0.315 | -0.078 | 0.008 |
| | d2q | $t_0$ | 1 | 0 | 0 | 0 | 0.111 | 0 | 0 | 0.342 | 0 | 0 | 0.297 | 0 | 0 |
| | | $t_1$ | 0.539 | 0.076 | 0.159 | 0.166 | 0.102* | 0.087 | -0.065 | 0.335 | 0.021 | -0.017 | 0.287 | 0.034 | 0.001 |
| | | $t_2$ | 0.552 | 0.075 | 0.151 | 0.160 | 0.123 | -0.109 | -0.004 | 0.374* | -0.093 | -0.091 | 0.327* | -0.101 | -0.015 |

aspects are contextualized in the literature and it is investigated which assumptions hold and which need to be further reviewed. Finally, limitations and directions for future work are outlined.

## 7.1 Evolving Document Rankings

Conventional test collection evaluations abstract the dynamics in a search environment. By considering multiple points in time in one evaluation, we systematically reintroduce dynamics in the search setup. The side-by-side comparison of the retrieval effectiveness shows that the results are not stable but fluctuate over time, sometimes drastically. Generally, it can be observed that the differences between EEs are larger if they span a longer period of time. This highlights the temporal connection between the sub-collections. While the LongEval test collection covers multiple months and shows minor changes, the TripClick test collection that simulates changes over decades shows stronger changes. Besides the time frame covered, the type and strength of changes influences how strong the ARP changes. Like expected, the experimental evaluations recommend that the effectiveness dependents on the EE. This observation shows that retrieval effectiveness evaluations based on the Cranfield paradigm are not temporally reliable by default.

The RBO and RMSE provide summarizing indicators for these observations as they measure the similarity between a progressed EE at $t_n$ and the initial EE at $t_0$. The more different the document ranking at a later point in time is, the lower the rank correlation, i.e., the lower the RBO scores are. The experimental evaluation supports this assumption in most investigated scenarios. For example, the RBO scores monotonically decrease for all systems on TripClick and TREC-COVID.

Based on these observations, we revisit the observation by Soboroff [44] that **"static test collections can be used to measure search in a changing document collection such as the live web […] measures such as bpref which work with incomplete information can be used with little or no additional relevance assessment."** In this original work, a web document collection (GOV2[4]) is investigated daily for a long period. We consider further search scenarios also from other domains and with additionl change types. The experimental results show that, indeed, bpref achieves more stable results on the change measures compared to MAP or P@10 in most progressed EEs. However, in the different scenarios influenced by the observed strong temporal effects, even the bpref scores varied. This highlights that the test collection dynamics also influence more robust measures like bpref. Further, the robustness of bpref comes at the cost of limited explainable power. Sakai [43] proposes adapted versions of MAP, nDCG, or the Q-measures as better alternatives to bpref. How the expressiveness of a measure relates to its temporal sensitivity remains an open question that may be investigated in future work. Soboroff's observation that measures like bpref can be used to assess the effectiveness in evolving document collections seems not to hold unrestricted. However, bpref seems to provide more persistent results.

## 7.2 Different Types of Change

Adar et al. [1] observed that **"some types of change are more meaningful"**. As stated before, how the results change over time

depends on the type and degree of changes in the collection. In the experimental evaluation, the presented search scenarios highlight the differences between various types of content with different change behaviors, such as general web pages (LongEval) and more specialized documents like academic papers or medical reports (TripClick and TREC-COVID). The results support the hypothesis that not all changes are equally important. This can be observed, for example, by comparing the raw collection statistics in Tab. 2 with the results from the change measures. This observation highlights the need for more precise quantifications of the changes in the test collections, for example, through measures like the Dice similarity [1].

The reproducibility measures can be interpreted as an extension of the test collection statistics as they can be seen as a surrogate of the changes that effectively impact the ranking. While the raw statistics in comparison might attest to drastic changes, e.g., half a million documents deleted as in LongEval $t_2$, the RBO and RMSE captures to what extent these changes actually affect the retrieval system. In this sense, the RBO and RMSE narrow down the changes in the test collection to the important ones that can impact the "actively consumed pages" as described by Adar et al. [1].

## 7.3 Comparing Effectiveness Across EEs

In addition to the reproducibility measures employed for the D'TQ scenarios, incorporating the measured effectiveness and additionally considering changes in the qrels potentially provides deeper insights and allows to investigate further scenarios. Since the measured effectiveness appears to be directly influenced by the EE, a direct comparison might not be meaningful and gained insights on how systems cope with these changes limited. Rather, if the effectiveness is directly compared, the system and the evolving EE are described simultaneously. This makes the scores difficult to compare and interpret since the differences in the qrels lead to a changed recall base. Sáez et al. [42] observed this for the $\mathcal{R}_e\Delta$ as **"$\mathcal{R}_e\Delta$: If the same IR system is evaluated in two EEs, extracting mainly the environmental effect on the system."**. They propose a pivot strategy that is essentially implemented in the $\Delta$RI.

The $\mathcal{R}_e\Delta$ directly describes the change in effectiveness compared to a previous point in time. Often, it appears to be connected to the dynamics of the test collections as summarized in Tab. 2. This suggests that the $\mathcal{R}_e\Delta$ is highly influenced by the changes in the EE, and the described initial observation seems to hold. In contrast to the $\mathcal{R}_e\Delta$, which is highly influenced by the changes in the EE, the $\Delta$RI seems to be able to dampen these influences by relating the measured effects to a pivot system. This should make the measured scores more comparable. It can be observed how the $\Delta$RI seem not to be related to the $\mathcal{R}_e\Delta$, especially in EE with overlapping change types. While the other measures show a strong agreement per EE, the $\Delta$RI rather agrees across systems.

How well the $\Delta$RI isolates the system effects needs to be further investigated in future work. A central question is the choice of the pivot system. Breuer et al. [8] choose it in relation to the advanced run by means that the advanced run should outperform the baseline run. Sáez et al. [42] investigate the choice of pivot system in the related temporal evaluation setting where also the systems change. They determine the quality of the pivot by its capability to rank the

---

[4]http://ir.dcs.gla.ac.uk/test_collections/gov2-summary.htm

experimental systems in the same order in different EEs. Compared to the applications in this work, different characteristics might be necessary.

## 7.4 Per Topic Changes

For test collection evaluations, it is established practice to average the effectiveness results across different topics. All topics are treated equally, although the effectiveness can vary potentially drastically between topics. While this is a known simplification, it is especially crucial to differentiate between topics if the temporal dynamics are assessed. Kulkarni et al. [29] observe: **"The content of documents also change with some documents always being relevant to a particular query and others being relevant to it at a particular point in time."**. This describes that the relevance changes over time, and that these changes can follow patterns.

Due to averaging across all topics, it remains difficult for most measures to uncover such dynamics if they overlay between topics. Like the ARP, the $\mathcal{R}_e\Delta$ and $\Delta$RI rely on effectiveness scores, averaged over all topics to describe the change in effectiveness over time. This leads to scenarios where, for example, good results on one topic compensate for weaker results on another topic. Since this compensation can follow patterns, it can be a structural problem, which makes it especially important to longitudinal evaluations.

Future work could regard identifying and accounting for such systematic errors in longitudinal evaluations. While these evaluations are only feasible on scenarios without changes in the topic component, a starting point is the RMSE that directly compares the effectiveness measured per topic. However, this comparison would require to compare effectiveness scores based on different qrel sets and, therefore, would be strongly influenced by the changing recall base. Potential measures that account for effectiveness changes per topic are suited to pick up on trends in the relevance labels. Measures that rely on the averaged results are more "forgiving" and rather describe the overall utility of the system.

## 7.5 Limitations and Future Work

Changes are manifold, and the results are limited in this respect. In the experimental evaluation, scenarios D'TQ and D'TQ' are examined, while many other and more specific scenarios are possible (e.g., Tab. 1). With increasing changes, not only in quantity but especially by considering multiple types of change in multiple components, the interpretation gets more complicated. For example, while the RBO only depends on changes in the document component, the $\Delta$RI and $\mathcal{R}_e\Delta$ also incorporates changes in the qrels. This fosters the need for principled experiments that distinguish the influences of different changes in different components on the retrieval effectiveness next to all possible combinations.

In our experiments, we notice changes in retrieval effectiveness but what does this actually mean for the users? The ARP scores and the derived relative measures like $\mathcal{R}_e\Delta$, RMSE, and others quantify the changes from a system-oriented point of view. Thinking about the users, we see that there are also changes in the rank correlations as measured by RBO. While there are performance drops at later points in time, these do not necessarily imply a lower rank correlation.

With regard to LongEval, we see lower ARP scores at $t_1$ than at $t_2$, while the RBO scores are very similar and sometimes have slightly contradictory scores, i.e., higher correlations but lower retrieval effectiveness. However, compared to $t_0$, the user experience will be different in both cases. This leaves us with the question what users will notice of the changing retrieval effectiveness. A more dramatic example is TREC-COVID. Here, we observe a steadily decreasing rank correlation along the different points in time, which could be explained by more substantial updates to the dataset over a shorter period. We note that our experiments only reveal a first glimpse of temporal changes from a strong system-oriented point of view. What kind of change is actually perceived by users remains as future work.

Further, the selection of datasets limits the interpretations of the results due to the sometimes extreme scenarios. While this may not always transfer to realistic or likely situations, the availability of test bends that consider temporal changes is rare. This strengthens the need for further temporal test collections and also strategies to simulate these.

## 8 CONCLUSION

In this work, we investigate the interplay between retrieval result, effectiveness and temporal changes in the evaluation setup represented as evaluation environment (EE). It was measured how retrieval systems cope with changes over time. We propose to classify the changes in the core components of a test collection: documents, topics, and qrels (DTQ) along the create, update, and delete operation known from CRUD. This gives differentiation to various search scenarios that are outlined.

Based on this conceptual model the influence of changing documents was investigated on the scenarios D'TQ where only the document component evolves and also in the D'TQ' scenario with additionally updated relevance judgments. We showed how known reproducibility measures can be adapted to quantify the influence on the retrieval results over time on different levels of granularity. The RBO summarizes the changes in the document collection that directly influence the effectiveness measures strictly based on the rankings. The RMSE similarly does so but allows that documents in the ranking are exchanged with equally relevant ones. The $\mathcal{R}_e\Delta$ directly describes the changing effectiveness. The $\Delta$RI also describes how the effectiveness changes but only after relating the effectiveness from both compared EEs to a pivot system what improves the comparability between stronger evolved EEs. The results of the experimental evaluation indicate how the effectiveness varies over time. This highlights that the results are not only dependent on the capabilities of the system but especially how strong they are influenced by the selected EE. We think this research is a valuable addition to temporal IR evaluations and thereby contributes to a more holistic understanding of IR evaluations in the ever-evolving information landscape.

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
