# OpenReview forum: "Evaluation of Temporal Change in IR Test Collections"
_ACM.org/SIGIR/ICTIR/2024/Conference — ICTIR 2024_

### Official Review · Reviewer_dEGx · 2024-05-13

**Rating:** 2
**Confidence:** 5

**Objective Part Of Review:**

This paper investigates the influence of temporal changes in test collections on retrieval results.
First, a general scheme for temporal developments is presented, distinguishing between documents, topics and qrels, with operations create, update and delete on each of them. Then different reproducibility measures are discussed wrt. the task at hand, and appropriate modifications are proposed.
Experiments are performed on three different testbeds, considering changes in documents only or in both documents and qrels. The temporal changes - especially wrt. to the retrieval methods compared - are bigger than expected.. These results are discussed in detail,
and followup research is proposed

**Subjective Part Of Review:**

The paper addresses an important topic that has found little attention in the past. The new scheme for characterizing temporal development is highly valuable for characterizing the presented (and future) research).
The paper presents the results of extensive experimentation, and the analysis of results is very thorough.
Overall, I think that this paper is a milestone in the investigation of temporal changes in IR testbeds.

---

### Official Review · Reviewer_rr3x · 2024-05-16

**Rating:** 1
**Confidence:** 4

**Objective Part Of Review:**

This paper presents an overview of initiatives dealing with evolving collections (documents, topics, and/or qrels). It considers three datasets (TripClick, TREC COVID and LongEval) and reports all the metrics that can compare systems between two versions of the dataset – it does also a good review of related works.

The result analysis (section 6) could be much improved. As it is, it is a mere description of the tables whereas it should try to show interesting facts (especially table 3 is quite big and unorganized, and would need to be analyzed in another way). At the end of the section, there is not much which has been learned.

The discussion (section 7: evolving document rankings / characterizing changes / comparing effectiveness / topic evolution) is much more interesting, and highlight important issues that should be already discussed in section 6 – which is a bit the case, but this aspect could be strengthened a lot. More importantly, those issues should also be more elaborated (e.g. "How well the $\Delta$RI isolates the system effects needs to be further investigated in future work" is really too vague).


Typos & questions:
- Why is monoBERT (or monoT5/etc.) not considered in the experiments? Since it is the model that behaves the best OOD, it would be interesting to have some insight on it.
- p.1 CRUD is not yet defined (and its usage not obvious before reading the remainder of the paper)
- eq. 1: there is a $j$ in the left-hand side of the equality... but none in the right
- eq. 2 (and others): $n^{EE}$ is not obvious as a notation for number of topics (unless I am wrong...)

**Subjective Part Of Review:**

The paper is easy to read and understand, and deals with a timely topic – that of continuous learning, evolving collections, which are quite important in an era of learned IR models.

The results in themselves could be interesting, but the paper fails to properly expose them and highlight important things: reading it, one wonders what is the real message and what are the important results that the authors report (section 6 and section 7 could be reworked).

Nonetheless, the paper is interesting for its overview of the different initiatives (collections, metrics) and its (short) discussion of current issues.

---

### Official Review · Reviewer_zHKN · 2024-05-17

**Rating:** 1
**Confidence:** 4

**Objective Part Of Review:**

This submission explores how to compare system effectiveness within an evaluation corpus that changes over time. I think the question is a great one to explore and I really love the idea of looking into it. I think that the way of describing how a corpus changes (e.g., CRUD) was clever and very easy to understand: nice. I also liked the detailed analysis, though as I'll note later, I wasn't quite sure what to make of it.

The related work seems to be missing some past work on corpora that "arrive" over time, sometimes called "streaming corpora." For example, the TREC Filtering track considered a constantly evolving collection of documents and a set of "standing" queries. The Topic Detection and Tracking (TDT) program had a similar streaming corpus and considered evaluation by looking a false alarms rather than precision, since the latter is a function of corpus richness (prior probability of relevance) and thus will change as the collection changes. The ways they handled a changing corpus are different from your goals, but they might suggest some additional baselines. At least they should probably not be ignored. (Both of those had their heyday 20+ years ago.)

Related to other baselines, I was wondering whether measures like InfAP and others that attempt to estimate average precision (etc.) with sampled or otherwise incomplete relevance assessments. Could they be adapted to this problem? Or, perhaps, do they already solve this problem?

I found the description of the evaluation measures difficult to follow. Mostly I think this was because of some small things that were confusing:

* In equation (1), isn't $1-\phi$ a constant (once $\phi$ is picked)? So what purpose does that have in the measure?
* When explaining what $A_i$ is in equation (1), I was (incorrectly) thinking it was overlap in relevant documents which caused confusion. Even if you think it's obvious, it might be worth tossing in a comment like "all documents, not just relevant ones" for people like me who were confused.
* Section 6.You describe something with high RMSE as having the "highest error." Although that's consistent with the name of the measure, you are really measuring the difference, not deviation from a correct truth. Perhaps just call it the "largest difference" or something to that effect?

**Subjective Part Of Review:**

As mentioned above, I was very drawn to this submission. In the end, though, it was not clear to me how I should interpret the results. Basically they show that "things change" but don't illuminate much beyond that. The submission provides a nice framework for thinking about change, but I feel that the evaluation measures are not  quite right or that the datasets are not ideal for exploring this question.

Here are a number of small points that created challenges from me. Most are trivial to fix and several are purely my own preferences -- that I obviously think you should change, but it's really just opinion.

* Abstract. I am unable to parse the sentence that starts "From the theoretical possible changes different"
* Although the world seems to be shifting away from this, normally "allows" requires a actor ("allows US to set") or a gerund ("allows setting"). Throws me, but as I say, I think "my side" is losing this one.
* Note that an apostrophe is not the same as \\$'\\$ in LaTeX.
* Section 3. reusable test collections are not just cost-efficient but also better for reproducibility.
* Abbreviating "Table" as "Tab" is atypical. Suggest not bothering to abbreviate it as it only saves the width of the letter e.
* Section 7.1. effectiveness dependents $\longrightarrow$ effectiveness depends
* Section 7.1. Usually a direct quotation from a paper requires a page number in the reference
* Section 7.3. "observed this for the ${\cal R}_e\Delta$ as "${\cal R}_e\Delta$: If the same IR system is evaluated in two EEs, extracting mainly the same environmental effect on the system." I cannot parse that. It either stops too soon or is missing something.
* Section 7.5, last paragraph. "availability of test bends that consider" - I cannot figure out what "bends" is supposed to be here.
* References [17] and [19] and [27] are incomplete.

---

### Official Review · Reviewer_w6Ah · 2024-05-20

**Rating:** 0
**Confidence:** 4

**Objective Part Of Review:**

The authors examine how retrieval results, effectiveness, and temporal changes within the evaluation setup—referred to as the evaluation environment (EE)—interact. The experimental setup can be represented as an evolving EE consisting of Documents, Topics, Qrels (DTQ).
 They suggest categorizing modifications in the core elements of a test collection (documents, topics, and qrels, abbreviated as DTQ) using the create, update, and delete operations from the CRUD model. They demonstrate that the proposed metrics effectively describe the changes in retrieval results.

through this they answer the question How can the impact of temporal changes in the evaluation setup on the retrieval results be quantified?  They used three test collections to simulate temporal effect. In addition, they used some  prior metrics to study the changes.  Similarly, five systems are also used. And this  work is interesting and relevant.

**Subjective Part Of Review:**

the notations are not introduced, for example, "We propose to classify the changes in the different components of Cranfield test collections by the create, update and delete operation of CRUD as high level differentiation." - what is CRUD here? it is not nice to second guess.
similarly, "In the first scenario, CREATE changes are explored." - again we need to second guess what create is?

not all measures introduced- bpref, It is common to introduce all metrics used for the evaluation and for consistency.

focusing on Figure 2- I am wondering what does this say? MonoT5 performance, on LongEval,  changes at three timepoints, why? Presumably, we want to know, is there any changes to system rank order? and if there is any changes in the rank order, we want to know why? This is not answered by the current analysis. Is the behaviour shown in Figure 2- expected or not? If the rank order of systems changed, then it may say the lack of robustness of the test set. Currently, the text just say what is observed in the figure without discussing the reasons.
"For example, the RBO scores monotonically decrease for all systems on TripClick and TREC-COVID." - the Rank Biased Overlap (RBO) compares two rankings of documents directly. for both TopClick and TREC-COVID you add documents and delete. Hence, it is possible that this measure is affected by this scenario

I kind of like the objectives of the study- Unfortunately, tehre is not much sensible explanation for the results observed.

---

### Meta-Review · Area_Chair_wdKu · 2024-05-31

**Recommendation:** Accept (Oral)
**Confidence:** 5

**Metareview:**

This paper presents a framework for evaluating search systems over a corpus that changes over time. The reviewers were unanimous in their appreciation for the topic of this paper, and I agree! Evolving corpora is a prevalent problem in real world search settings and the presented framework is a very tidy formulation, and already provides some interesting insights.

The reviewers noted that improvements to the presentation would make the paper stronger, and provided detailed feedback. Please read them carefully and incorporate their suggestions, especially w.r.t. better highlighting key insights and takeaways. I agree that Sections 6 & 7 could use a reworking and Table 3 is quite the monster table!

Reviewer zHKN also notes some relevant past TREC tracks that the authors may find relevant and consider citing in this paper.